# Learning Nonsymmetric Determinantal Point Processes

**Mike Gartrell**
Criteo AI Lab
m.gartrell@criteo.com

**Victor-Emmanuel Brunel**
ENSAE ParisTech
victor.emmanuel.brunel@ensae.fr

**Elvis Dohmatob**
Criteo AI Lab
e.dohmatob@criteo.com

**Syrine Krichene** *
Criteo AI Lab
syrinekrichene@google.com

## Abstract

Determinantal point processes (DPPs) have attracted substantial attention as an elegant probabilistic model that captures the balance between quality and diversity within sets. DPPs are conventionally parameterized by a positive semi-definite kernel matrix, and this symmetric kernel encodes only repulsive interactions between items. These so-called symmetric DPPs have significant expressive power, and have been successfully applied to a variety of machine learning tasks, including recommendation systems, information retrieval, and automatic summarization, among many others. Efficient algorithms for learning symmetric DPPs and sampling from these models have been reasonably well studied. However, relatively little attention has been given to nonsymmetric DPPs, which relax the symmetric constraint on the kernel. Nonsymmetric DPPs allow for both repulsive and *attractive* item interactions, which can significantly improve modeling power, resulting in a model that may better fit for some applications. We present a method that enables a tractable algorithm, based on maximum likelihood estimation, for learning nonsymmetric DPPs from data composed of observed subsets. Our method imposes a particular decomposition of the nonsymmetric kernel that enables such tractable learning algorithms, which we analyze both theoretically and experimentally. We evaluate our model on synthetic and real-world datasets, demonstrating improved predictive performance compared to symmetric DPPs, which have previously shown strong performance on modeling tasks associated with these datasets.

## 1 Introduction

Determinantal point processes (DPPs) have attracted growing attention from the machine learning community as an elegant probablistic model for the relationship between items within observed subsets, drawn from a large collection of items. DPPs have been well studied for their theoretical properties [1, 4, 9, 13, 18, 20, 21], and have been applied to numerous machine learning applications, including document summarization [7, 24], recommender systems [11], object retrieval [1], sensor placement [17], information retrieval [19], and minibatch selection [29]. Efficient algorithms for DPP learning [10, 12, 14, 25, 26] and sampling [2, 22, 27] have been reasonably well studied. DPPs are conventionally parameterized by a positive semi-definite (PSD) kernel matrix, and due to this symmetric kernel, they are able to encode only repulsive interactions between items. Despite this limitation, symmetric DPPs have significant expressive power, and have proven effective in the aforementioned applications. However, the ability to encode only repulsive interactions, or negative

correlations between pairs of items, does have important limitations in some settings. For example, consider the case of a recommender system for a shopping website, where the task is to provide good recommendations for items to complete a user's shopping basket prior to checkout. For models that can only encode negative correlations, such as the symmetric DPP, it is impossible to directly encode positive interactions between items; e.g., a purchased basket containing a video game console would be more likely to also contain a game controller. One way to resolve this limitation is to consider nonsymmetric DPPs, which relax the symmetric constraint on the kernel.

Nonsymmetric DPPs allow the model to encode both repulsive and *attractive* item interactions, which can significantly improve modeling power. With one notable exception [5], little attention has been given to nonsymmetric DPPs within the machine learning community. We present a method for learning fully nonsymmetric DPP kernels from data composed of observed subsets, where we leverage a low-rank decomposition of the nonsymmetric kernel that enables a tractable learning algorithm based on maximum likelihood estimation (MLE).

**Contributions** Our work makes the following contributions:

• We present a decomposition of the nonsymmetric DPP kernel that enables a tractable MLE-based learning algorithm. To the best of our knowledge, this is the first MLE-based learning algorithm for nonsymmetric DPPs.
• We present a general framework for the theoretical analysis of the properties of the maximum likelihood estimator for a somewhat restricted class of nonsymmetric DPPs, which shows that this estimator has particular statistical guarantees regarding consistency.
• Through an extensive experimental evaluation on several synthetic and real-world datasets, we highlight the significant improvements in modeling power that nonsymmetric DPPs provide in comparison to symmetric DPPs. We see that nonsymmetric DPPs are more effective at recovering correlation structure within data, particularly for data that contains large disjoint collections of items.

Unlike previous work on signed DPPs [5], our work does not make the very limiting assumption that the correlation kernel of the DPP is symmetric in the absolute values of the entries. This gives our model much more flexibility. Moreover, our learning algorithm, based on maximum likelihood estimation, allows us to leverage a low rank assumption on the kernel, while the method of moments tackled in [5] does not seem to. Finally, the learning algorithm in [5] has computational complexity of $O(M^6)$, where $M$ is the size of the ground set (e.g., item catalog), making it computationally infeasible for most practical scenarios. In contrast, our learning algorithm has substantially lower time complexity of $O(M^3)$, which allows our approach to be used on many real-world datasets.

## 2    Background

A DPP models a distribution over subsets of a finite ground set $\mathcal{Y}$ that is parametrized by a matrix $\boldsymbol{L} \in \mathbb{R}^{|\mathcal{Y}| \times |\mathcal{Y}|}$, such that for any $J \subseteq \mathcal{Y}$,

$$\Pr(J) \propto \det(\boldsymbol{L}_J),\tag{1}$$

where $\boldsymbol{L}_J = [\boldsymbol{L}_{ij}]_{i,j \in J}$ is the submatrix of $\boldsymbol{L}$ indexed by $J$.

Since the normalization constant for Eq. 1 follows from the observation that $\sum_{J \subseteq \mathcal{Y}} \det(\boldsymbol{L}_J) = \det(\boldsymbol{L} + \boldsymbol{I})$, we have, for all $J \subseteq \mathcal{Y}$,

$$\mathcal{P}_{\boldsymbol{L}}(J) = \frac{\det(\boldsymbol{L}_J)}{\det(\boldsymbol{L} + \boldsymbol{I})}.\tag{2}$$

Without loss of generality, we will assume that $\mathcal{Y} = \{1, 2, \ldots, M\}$, which we also denote by $[M]$, where $M \geq 1$ is the cardinality of $\mathcal{Y}$.

It is common to assume that $\boldsymbol{L}$ is a positive semi-definite matrix in order to ensure that $\mathcal{P}_{\boldsymbol{L}}$ defines a probability distribution on the power set of $[M]$ [20]. More generally, any matrix $\boldsymbol{L}$ whose principal minors $\det(\boldsymbol{L}_J), J \subseteq [M]$, are nonnegative, is admissible to define a probability distribution as in (2) [5]; such matrices are called $P_0$-matrices. Recall that any matrix $\boldsymbol{L}$ can be decomposed uniquely as the sum of a symmetric matrix $\boldsymbol{S}$ and a skew-symmetric matrix $\boldsymbol{A}$. Namely, $\boldsymbol{S} = \frac{\boldsymbol{L} + \boldsymbol{L}^\top}{2}$ whereas $\boldsymbol{A} = \frac{\boldsymbol{L} - \boldsymbol{L}^\top}{2}$. The following lemma gives a simple sufficient condition on $\boldsymbol{S}$ for $\boldsymbol{L}$ to be a $P_0$-matrix.

**Lemma 1.** *Let $\boldsymbol{L} \in \boldsymbol{R}^{M \times M}$ be an arbitrary matrix. If $\boldsymbol{L} + \boldsymbol{L}^\top$ is PSD, then $\boldsymbol{L}$ is a $P_0$-matrix.*

An important consequence is that a matrix of the form $\boldsymbol{D} + \boldsymbol{A}$, where $\boldsymbol{D}$ is diagonal with positive diagonal entries and $\boldsymbol{A}$ is skew-symmetric, is a $P_0$-matrix. Such a matrix would only capture nonnegative correlations, as explained in the next section.

## 2.1 Capturing Positive and Negative Correlations

When DPPs are used to model real data, they are often formulated in terms of the $\boldsymbol{L}$ matrix as described above, called an $\boldsymbol{L}$-ensemble. However, DPPs can be alternatively represented in terms of the $M \times M$ matrix $\boldsymbol{K}$, where $\boldsymbol{K} = \boldsymbol{I} - (\boldsymbol{L} + \boldsymbol{I})^{-1}$. Using the $\boldsymbol{K}$ representation,

$$\Pr(J \subseteq Y) = \det(\boldsymbol{K}_J), \tag{3}$$

where $Y$ is a random subset drawn from $\mathcal{P}$. $\boldsymbol{K}$ is called the marginal kernel; since here we are defining marginal probabilities that don't need to sum to 1, no normalization constant is needed. DPPs are conventionally parameterized by a PSD $\boldsymbol{K}$ or $\boldsymbol{L}$ matrix, which is symmetric.

However, $\boldsymbol{K}$ and $\boldsymbol{L}$ need not be symmetric. As shown in [5], $\boldsymbol{K}$ is admissible if and only if $\boldsymbol{L}$ is a $P_0$ matrix, that is, all of its principal minors are nonnegative. The class of $P_0$ matrices is much larger, and allows us to accommodate nonsymmetric $\boldsymbol{K}$ and $\boldsymbol{L}$ matrices. To enforce the $P_0$ constraint on $\boldsymbol{L}$ during learning, we impose the decomposition of $\boldsymbol{L}$ described in Section 4. Since we see as consequence of Lemma 1 that the sum of a PSD matrix and a skew-symmetric matrix is a $P_0$ matrix, this allows us to support nonsymmetric kernels, while ensuring that $\boldsymbol{L}$ is a $P_0$ matrix. As we will see in the following, there are significant advantages to accommodating nonsymmetric kernels in terms of modeling power.

As shown in [20], the eigenvalues of $\boldsymbol{K}$ are bounded above by one, while $\boldsymbol{L}$ need only be a PSD or $P_0$ matrix. Furthermore, $\boldsymbol{K}$ gives the marginal probabilities of subsets, while $\boldsymbol{L}$ directly models the atomic probabilities of observed each subset of $\mathcal{Y}$. For these reasons, most work on learning DPPs from data uses the $\boldsymbol{L}$ representation of a DPP.

If $J = \{i\}$ is a singleton set, then $\Pr(i \in Y) = K_{ii}$. The diagonal entries of $\boldsymbol{K}$ directly correspond to the marginal inclusion probabilities for each element of $\mathcal{Y}$. If $J = \{i, j\}$ is a set containing two elements, then we have

$$\Pr(i, j \in Y) = \begin{vmatrix} K_{ii} & K_{ij} \\ K_{ji} & K_{jj} \end{vmatrix} = K_{ii}K_{jj} - K_{ij}K_{ji}. \tag{4}$$

Therefore, the off-diagonal elements determine the correlations between pairs of items; that is, $\mathrm{cov}(\mathbb{1}_{i \in Y}, \mathbb{1}_{j \in Y}) = -K_{ij}K_{ji}$. For a symmetric $\boldsymbol{K}$, the signs and magnitudes of $K_{ij}$ and $K_{ji}$ are the same, resulting in $\mathrm{cov}(\mathbb{1}_{i \in Y}, \mathbb{1}_{j \in Y}) = -K_{ij}^2 \leq 0$. We see that in this case, the off-diagonal elements represent negative correlations between pairs of items, where a larger value of $K_{ij}$ leads to a lower probability of $i$ and $j$ co-occurring, while a smaller value of $K_{ij}$ indicates a higher co-occurrence probability. If $K_{ij} = 0$, then there is no correlation between this pair of items. Since the sign of the $-K_{ij}^2$ term is always nonpositive, the symmetric model is able to capture only nonpositive correlations between items. In fact, symmetric DPPs induce a strong negative dependence between items, called negative association [3].

For a nonsymmetric $\boldsymbol{K}$, the signs and magnitudes of $K_{ij}$ and $K_{ji}$ may differ, resulting in $\mathrm{cov}(\mathbb{1}_{i \in Y}, \mathbb{1}_{j \in Y}) = -K_{ij}K_{ji} \geq 0$. In this case, the off-diagonal elements represent positive correlations between pairs of items, where a larger value of $K_{ij}K_{ji}$ leads to a higher probability of $i$ and $j$ co-occurring, while a smaller value of $K_{ij}K_{ji}$ indicates a lower co-occurrence probability. Of course, the signs of the off-diagonal elements for some pairs $(i, j)$ may be the same in a nonsymmetric $\boldsymbol{K}$, which allows the model to also capture negative correlations. Therefore, a nonsymmetric $\boldsymbol{K}$ can capture both negative and positive correlations between pairs of items.

## 3 General guarantees in maximum likelihood estimation for DPPs

In this section we define the log-likelihood function and we study the Fisher information of the model. The Fisher information controls whether the maximum likelihood, computed on $n$ iid samples, will be a $\sqrt{n}$-consistent. When the matrix $\boldsymbol{L}$ is not invertible (i.e., if it is only a $P_0$-matrix and not a $P$-matrix), the support of $\mathcal{P}_L$, defined as the collection of all subsets $J \subseteq [M]$ such that $\mathcal{P}_{\boldsymbol{L}}(J) \neq 0$, depends on $\boldsymbol{L}$, and the Fisher information will not be defined in general. Hence, we will assume, in this section, that $\boldsymbol{L}$ is invertible and that we only maximize the log-likelihood over classes of invertible matrices $\boldsymbol{L}$.

Consider a subset $\Theta$ of the set of all $P$-matrices of size $M$. Given a collection of $n$ observed subsets $\{Y_1, ..., Y_n\}$ composed of items from $\mathcal{Y} = [M]$, our learning task is to fit a DPP kernel $\boldsymbol{L}$ based on this data. For all $\boldsymbol{L} \in \Theta$, the log-likelihood is defined as

$$\hat{f}_n(\boldsymbol{L}) = \frac{1}{n} \sum_{i=1}^{n} \log \mathcal{P}_{\boldsymbol{L}}(Y_i) = \sum_{J \subseteq [M]} \hat{p}_J \log \mathcal{P}_{\boldsymbol{L}}(J) = \sum_{J \subseteq [M]} \hat{p}_J \log \det(\boldsymbol{L}_J) - \log \det(\boldsymbol{L} + \boldsymbol{I})$$

(5)

where $\hat{p}_J$ is the proportion of observed samples that equal $J$.

Now, assume that $Y_1, \ldots, Y_n$ are iid copies of a DPP with kernel $\boldsymbol{L}^* \in \Theta$. For all $\boldsymbol{L} \in \Theta$, the population log-likelihood is defined as the expectation of $\hat{f}_n(\boldsymbol{L})$, i.e.,

$$f(\boldsymbol{L}) = \mathbb{E}\left[\log \mathcal{P}_{\boldsymbol{L}}(Y_1)\right] = \sum_{J \subseteq [M]} p_J^* \log \det(\boldsymbol{L}_J) - \log \det(\boldsymbol{L} + \boldsymbol{I})$$

(6)

where $p_J^* = \mathbb{E}[\hat{p}_J] = \log \mathcal{P}_{\boldsymbol{L}^*}(J)$.

The maximum likelihood estimator (MLE) is defined as a minimizer $\hat{\boldsymbol{L}}$ of $\hat{f}_n(\boldsymbol{L})$ over the parameter space $\Theta$. Since $\hat{\boldsymbol{L}}$ can be viewed as a perturbed version of $\boldsymbol{L}^*$, it can be convenient to introduce the space $\mathcal{H}$ defined as the linear subspace of $\boldsymbol{R}^{M \times M}$ spanned by $\Theta$ and define the successive derivatives of $\hat{f}_n$ and $f$ as multilinear forms on $\mathcal{H}$. As we will see later on, the complexity of the model can be captured in the size of the space $\mathcal{H}$. The following lemma provides a few examples. We say that a matrix $\boldsymbol{L}$ is a *signed matrix* if for all $i \neq j$, $L_{i,j} = \varepsilon_{i,j} L_{j,i}$ for some $\varepsilon_{i,j} \in \{-1, 1\}$.

**Lemma 2.** *1. If $\Theta$ is the set of all positive definite matrices, it is easy to see that $\mathcal{H}$ is the set of all symmetric matrices.*

*2. If $\Theta$ is the set of all $P$-matrices, then $\mathcal{H} = \boldsymbol{R}^{M \times M}$.*

*3. If $\Theta$ is the collection of signed $P$-matrices, then $\mathcal{H} = \boldsymbol{R}^{M \times M}$.*

*4. If $\Theta$ is the set of $P$-matrices of the form $\boldsymbol{S} + \boldsymbol{A}$, where $\boldsymbol{S}$ is a symmetric matrix and $\boldsymbol{A}$ is a skew-symmetric matrix (i.e., $\boldsymbol{A}^\top = -\boldsymbol{A}$), then $\mathcal{H} = \boldsymbol{R}^{M \times M}$.*

*5. If $\Theta$ is the set of signed $P$-matrices with known signed pattern (i.e., there exists $(\varepsilon_{i,j})_{i>j} \subseteq \{-1, 1\}$ such that for all $\boldsymbol{L} \in \Theta$ and all $i > j$, $L_{j,i} = \varepsilon_{i,j} L_{i,j}$), then $\mathcal{H}$ is the collection of all signed matrices with that same sign pattern. In particular, if $\Theta$ is the set of all $P$-matrices of the form $\boldsymbol{D} + \boldsymbol{A}$ where $\boldsymbol{D}$ is diagonal and $\boldsymbol{A}$ is skew-symmetric, then $\mathcal{H}$ is the collection of all matrices that are the sum of a diagonal and a skew-symmetric matrix.*

It is easy to see that the population log-likelihood $f$ is infinitely many times differentiable on the relative interior of $\Theta$ and that for all $L$ in the relative interior of $\Theta$ and $\boldsymbol{H} \in \mathcal{H}$,

$$\mathrm{d}f(\boldsymbol{L})(\boldsymbol{H}) = \sum_{J \subseteq [M]} p_J^* \operatorname{tr}\left(\boldsymbol{L}_J^{-1} \boldsymbol{H}_J\right) - \operatorname{tr}\left((\boldsymbol{I} + \boldsymbol{L})^{-1} \boldsymbol{H}\right)$$

(7)

and

$$\mathrm{d}^2 f(\boldsymbol{L})(\boldsymbol{H}, \boldsymbol{H}) = -\sum_{J \subseteq [M]} p_J^* \operatorname{tr}\left((\boldsymbol{L}_J^{-1} \boldsymbol{H}_J)^2\right) + \operatorname{tr}\left(\left((\boldsymbol{I} + \boldsymbol{L})^{-1} \boldsymbol{H}\right)^2\right).$$

(8)

Hence, we have the following theorem. The case of symmetric kernels is studied in [6] and the following result is a straightforward extension to arbitrary parameter spaces. For completeness, we include the proof in the appendix. For a set $\Theta \subseteq \boldsymbol{R}^{N \times N}$, we call the relative interior of $\Theta$ its interior in the linear space spanned by $\Theta$.

**Theorem 1.** *Let $\Theta$ be a set of $P$-matrices and let $\boldsymbol{L}^*$ be in the relative interior of $\Theta$. Then, for all $\boldsymbol{H} \in \mathcal{H}$, $\mathrm{d}f(\boldsymbol{L}^*)(\boldsymbol{H}) = 0$. Moreover, the Fisher information is the negative Hessian of $f$ at $\boldsymbol{L}^*$ and is given by*

$$-\mathrm{d}^2 f(\boldsymbol{L}^*)(\boldsymbol{H}, \boldsymbol{H}) = \mathit{Var}\operatorname{tr}\left((\boldsymbol{L}_Y^*)^{-1} \boldsymbol{H}_Y\right),$$

(9)

*where $Y$ is a DPP with kernel $\boldsymbol{L}^*$.*

It follows that the Fisher information is positive definite if and only if any $\boldsymbol{H} \in \mathcal{H}$ that verifies

$$\operatorname{tr}\left((\boldsymbol{L}_J^*)^{-1} \boldsymbol{H}_J\right) = 0, \forall J \subseteq [M]$$

(10)

must be $H = 0$. When $\Theta$ is the space of symmetric and positive definite kernels, the Fisher information is definite if and only if $L^*$ is irreducible, i.e., it is not block-diagonal up to a permutation of its rows and columns [6]. In that case, it is shown that the MLE learns $\boldsymbol{L}^*$ at the speed $n^{-1/2}$. In general, this property fails and even irreducible kernels can induce a singular Fisher information.

**Lemma 3.** *Let $\Theta$ be a subset of $\boldsymbol{P}$-matrices.*

*1. Let $\boldsymbol{L}^* \in \Theta$ and $\boldsymbol{H} \in \mathcal{H}$ satisfy (10). Then, for all $i \in [M]$, $H_{i,i} = 0$.*

*2. Let $i, j \in [M]$ with $i \neq j$. Let $\boldsymbol{L}^* \in \Theta$ be such that $L_{i,j}^* \neq 0$ and $\mathcal{H}$ satisfy the following property: $\exists \varepsilon \neq 0$ such that $\forall \boldsymbol{H} \in \mathcal{H}$, $H_{j,i} = \varepsilon H_{i,j}$. Then, if $\boldsymbol{H} \in \mathcal{H}$ satisfies (10), $H_{i,j} = H_{j,i} = 0$.*

*3. Let $\boldsymbol{L}^* \in \Theta$ be block diagonal. Then, any $\boldsymbol{H} \in \mathcal{H}$ supported outside of the diagonal blocks of $\boldsymbol{L}^*$ satisfies (10).*

In particular, this lemma implies that if $\Theta$ is a class of signed $P$-matrices with prescribed sign pattern (i.e., $L_{i,j} = \varepsilon_{i,j} L_{j,i}$ for all $i \neq j$ and all $\boldsymbol{L} \in \Theta$, where the $\varepsilon_{i,j}$'s are $\pm 1$ and do not depend on $\boldsymbol{L}$), then if $\boldsymbol{L}^*$ lies in the relative interior of $\Theta$ and has no zero entries, the Fisher information is definite.

In the symmetric case, it is shown in [6] that the only matrices $\boldsymbol{H}$ satisfying (10) must be supported off the diagonal blocks of $\boldsymbol{L}^*$, i.e., the third part of Lemma 3 is an equivalence. In the appendix, we provide a few very simple counterexamples that show that this equivalence is no longer valid in the nonsymmetric case.

## 4 Model

To add support for positive correlations to the DPP, we consider nonsymmetric $\boldsymbol{L}$ matrices. In particular, our approach involves incorporating a skew-symmetric perturbation to the PSD $\boldsymbol{L}$.

Recall that any matrix $\boldsymbol{L}$ can be uniquely decomposed as $\boldsymbol{L} = \boldsymbol{S} + \boldsymbol{A}$, where $\boldsymbol{S}$ is symmetric and $\boldsymbol{A}$ is skew-symmetric. We impose a decomposition on $\boldsymbol{A}$ as $\boldsymbol{A} = \boldsymbol{B}\boldsymbol{C}^T - \boldsymbol{C}\boldsymbol{B}^T$, where $\boldsymbol{B}$ and $\boldsymbol{C}$ are low-rank $M \times D'$ matrices, and we use a low-rank factorization of $\boldsymbol{S}$, $\boldsymbol{S} = \boldsymbol{V}\boldsymbol{V}^T$, where $\boldsymbol{V}$ is a low-rank $M \times D$ matrix, as described in [12], which also allows us to enforce $\boldsymbol{S}$ to be PSD and hence, $\boldsymbol{L}$ to be a $P_0$-matrix by Lemma 1.

We define a regularization term, $R(\boldsymbol{V}, \boldsymbol{B}, \boldsymbol{C})$, as

$$R(\boldsymbol{V}, \boldsymbol{B}, \boldsymbol{C}) = -\alpha \sum_{i=1}^{M} \frac{1}{\lambda_i} \|\boldsymbol{v}_i\|_2^2 - \beta \sum_{i=1}^{M} \frac{1}{\lambda_i} \|\boldsymbol{b}_i\|_2^2 - \gamma \sum_{i=1}^{M} \frac{1}{\lambda_i} \|\boldsymbol{c}_i\|_2^2 \tag{11}$$

where $\lambda_i$ counts the number of occurrences of item $i$ in the training set, $\boldsymbol{v}_i$, $\boldsymbol{b}_i$, and $\boldsymbol{c}_i$ are the corresponding row vectors of $\boldsymbol{V}$, $\boldsymbol{B}$, and $\boldsymbol{C}$, respectively, and $\alpha, \beta, \gamma > 0$ are tunable hyperparameters. This regularization formulation is similar to that proposed in [12]. From the above, we have the full formulation of the regularized log-likelihood of our model:

$$\phi(\boldsymbol{V}, \boldsymbol{B}, \boldsymbol{C}) = \sum_{i=1}^{n} \log \det \left( \boldsymbol{V}_{Y_i} \boldsymbol{V}_{Y_i}^T + (\boldsymbol{B}_{Y_i} \boldsymbol{C}_{Y_i}^T - \boldsymbol{C}_{Y_i} \boldsymbol{B}_{Y_i}^T) \right) - \log \det \left( \boldsymbol{V}\boldsymbol{V}^T + (\boldsymbol{B}\boldsymbol{C}^T - \boldsymbol{C}\boldsymbol{B}^T) + \boldsymbol{I} \right)$$
$$+ R(\boldsymbol{V}, \boldsymbol{B}, \boldsymbol{C}) \tag{12}$$

The computational complexity of Eq. 12 will be dominated by computing the determinant in the second term (the normalization constant), which is $O(M^3)$. Furthermore, since $\frac{\partial}{\partial L_{ij}}(\log \det(\boldsymbol{L})) = \text{tr}(\boldsymbol{L}^{-1} \frac{\partial \boldsymbol{L}}{\partial L_{ij}})$, the computational complexity of computing the gradient of Eq. 12 during learning will be dominated by computing the matrix inverse in the gradient of the second term, $(\boldsymbol{L} + \boldsymbol{I})^{-1}$, which is $O(M^3)$. Therefore, we see that the low-rank decomposition of the kernel in our nonsymmetric model does not afford any improvement over a full-rank model in terms of computational complexity. However, our low-rank decomposition does provide a savings in terms of the memory required to store model parameters, since our low-rank model has space complexity $O(MD + 2MD')$, while a full-rank version of this nonsymmetric model has space complexity $O(M^2 + 2M^2)$. When $D \ll M$ and $D' \ll M$, which is typical in many settings, this will result in a significant space savings.

## 5 Experiments

We run extensive experiments on several synthetic and real-world datasets. Since the focus of our work is on improving DPP modeling power and comparing nonsymmetric and symmetric DPPs, we use the standard symmetric low-rank DPP as the baseline model for our experiments.

**Preventing numerical instabilities** The first term on the right side of Eq. (12) will be singular whenever $|Y_i| > D$, where $Y_i$ is an observed subset. Therefore, to address this in practice we set $D$

to the size of the largest subset observed in the data, as explained in [12]. Furthermore, the first term on the right side of Eq. (12) may be singular even when $|Y_i| \leq D$. In this case, we know that we are not at a maximum, since the value of the function becomes $-\infty$. Numerically, to prevent such singularities, in our implementation we add a small $\epsilon I$ correction to each $L_{Y_i}$ when optimizing Eq. 12 (we set $\epsilon = 10^{-5}$ in our experiments).

## 5.1 Datasets

We perform next-item prediction and AUC-based classification experiments on two real-world datasets composed of purchased shopping baskets:

1. **Amazon Baby Registries:** This public dataset consists of 111,0006 registries or "baskets" of baby products, and has been used in prior work on DPP learning [11, 14, 25]. The registries are collected from 15 different categories, such as "apparel", "diapers", etc., and the items in each category are disjoint. We evaluate our models on the popular apparel category.

We also perform an evaluation on a dataset composed of the three most popular categories: apparel, diaper, and feeding. We construct this dataset, composed of three large disjoint categories of items, with a catalog of 100 items in each category, to highlight the differences in how nonsymmetric and symmetric DPPs model data. In particular, we will see that the nonsymmetric DPP uses positive correlations to capture item co-occurences within baskets, while negative correlations are used to capture disjoint pairs of items. In contrast, since symmetric DPPs can only represent negative correlations, they must attempt to capture both co-occuring items, and items that are disjoint, using only negative correlations.

2. **UK Retail:** This is a public dataset [8] that contains 25,898 baskets drawn from a catalog of 4,070 items. This dataset contains transactions from a non-store online retail company that primarily sells unique all-occasion gifts, and many customers are wholesalers. We omit all baskets with more than 100 items, which allows us to use a low-rank factorization of the symmetric DPP ($D = 100$) that scales well in training and prediction time, while also keeping memory consumption for model parameters to a manageable level.

3. We also perform an evaluation on synthetically generated data. Our data generator allows us to explicitly control the item catalog size, the distribution of set sizes, and the item co-occurrence distribution. By controlling these parameters, we are able to empirically study how the nonsymmetric and symmetric models behave for data with a specified correlation structure.

## 5.2 Experimental setup and metrics

Next-item prediction involves identifying the best item to add to a subset of selected items (e.g., basket completion), and is the primary prediction task we evaluate.

We compute a next-item prediction for a basket $J$ by conditioning the DPP on the event that all items in $J$ are observed. As described in [13], we compute this conditional kernel, $L^J$, as $L^J = L_{\bar{J}} - L_{\bar{J},J} L_J^{-1} L_{J,\bar{J}}$, where $\bar{J} = \mathcal{Y} - J$, $L_{\bar{J}}$ is the restriction of $L$ to the rows and columns indexed by $\bar{J}$, and $L_{\bar{J},J}$ consists of the $\bar{J}$ rows and $J$ columns of $L$. The computational complexity of this operation is dominated by the three matrix multiplications, which is $O(M^2|J|)$.

We compare the performance of all methods using a standard recommender system metric: mean percentile rank (MPR). A MPR of 50 is equivalent to random selection; a MPR of 100 indicates that the model perfectly predicts the held out item. MPR is a recall-based metric which we use to evaluate the model's predictive power by measuring how well it predicts the next item in a basket; it is a standard choice for recommender systems [15, 23]. See Appendix C for a formal description of how the MPR metric is computed.

We evaluate the discriminative power of each model using the AUC metric. For this task, we generate a set of negative subsets uniformly at random. For each positive subset $J^+$ in the test set, we generate a negative subset $J^-$ of the same length by drawing $|J^+|$ samples uniformly at random, and ensure that the same item is not drawn more than once for a subset. We compute the AUC for the model on these positive and negative subsets, where the score for each subset is the log-likelihood that the model assigns to the subset. This task measures the ability of the model to discriminate between observed positive subsets (ground-truth subsets) and randomly generated subsets.

For all experiments, a random selection of 80% of the baskets are used for training, and the remaining 20% are used for testing. We use a small held-out validation set for tracking convergence and tuning

hyperparameters. Convergence is reached during training when the relative change in validation log-likelihood is below a pre-determined threshold, which is set identically for all models. We implement our models using PyTorch [2], and use the Adam [16] optimization algorithm to train our models.

## 5.3 Results on synthetic datasets

We run a series of synthetic experiments to examine the differences between nonsymmetric and symmetric DPPs. In all of these experiments, we define an oracle that controls the generative process for the data. The oracle uses a deterministic policy to generate a dataset composed of positive baskets (items that co-occur) and negative baskets (items that don't co-occur). This generative policy defines the expected normalized determinant, $\det(\boldsymbol{K}_J)$, for each pair of items, and a threshold that limits the maximum determinantal volume for a positive basket and the minimum volume for a negative basket. This threshold is used to compute AUC results for this set of positives and negatives. Note that the negative sets are used only during evaluation. For each experiment, in Figures 1, 2, and 3, we plot a transformed version of the learned $\boldsymbol{K}$ matrices for the nonsymmetric and symmetric models, where each element $i$ of this matrix is re-weighted by $\det(\boldsymbol{K}_{\{ij\}})$ for the corresponding pair. For each plotted transformation of $\boldsymbol{K}$, a magenta element corresponds to a negative correlation, which will tend to result in the model predicting that the corresponding pair is negative pair. Black and cyan elements correspond to smaller and larger positive correlations, respectively, for the nonsymmetric model, and very small negative correlations for the symmetric model; the model will tend to predict that the corresponding pair is positive in these cases. We perform the AUC-based evaluation for each pair $\{i, j\}$ by comparing $\det(\boldsymbol{K}_{\{ij\}})$ predicted by the model with the ground truth determinantal volume provided by the oracle; this task is equivalent to performing basket completion for the pair. In Figures 1, 2, and 3, we show the prediction error for each pair, where cyan corresponds to low error, and magenta corresponds to high error.

**Recovering positive examples for low-sparsity data**  In this experiment we aim to show that the nonsymmetric model is just as capable as the symmetric model when it comes to learning negative correlations when trained on data containing few negative correlations and many positive correlations. We choose a setting where the symmetric model performs well. We construct a dataset that contains no large disjoint collections of items, with 100 baskets of size six, and a catalog of 100 items. To reduce the impact of negative correlations between items, we use a categorical distribution, with nonuniform event probabilities, for sampling the items that populate each basket, with a large coverage of possible item pairs. This logic ensures few negative correlations, since there is a low probability that two items will never co-occur. For the nonsymmetric DPP, the oracle expects the model to predict a low negative correlation, or a positive correlation, for a pair of products that have a high co-occurence probability in the data. The results of this experiment are shown in Figure 1. We see from the plots showing the transformed $\boldsymbol{K}$ matrices that both the nonsymmetric and symmetric models recover approximately the same structure, resulting in similar error plots, and similar predictive AUC of approximately 0.8 for both models.

**Recovering negative examples for high-sparsity data**  We construct a more challenging scenario for this experiment, which reveals an important limitation of the symmetric DPP. The symmetric DPP requires a relatively high density of observed item pairs (positive pairs) in order to learn the negative structure of the data that describes items that do not co-occur. During learning, the DPP will maximize determinantal volumes for positive pairs, while the $\det(\boldsymbol{L} + \boldsymbol{I})$ normalization constant maintains a representation of the global volume of the parameter space for the entire item catalog. For a high density of observed positive pairs, increasing the volume allocated to positive pairs will result in a decrease in the volume assigned to many negative pairs, in order to maintain approximately the same global volume represented by the normalization constant. For a low density of positive pairs, the model will not allocate low volumes to many negative pairs. This phenomenon affects both the nonsymmetric and symmetric models. Therefore, the difference in each model's ability to capture negative structure within a low-density region of positive can be explained in terms of how each model maximizes determinatal volumes using positive and negative correlations. In the case of the symmetric DPP, the model can increase determinantal volumes by using smaller negative correlations, resulting in off-diagonal $K_{ij} = K_{ji}$ values that approach zero. As these off-diagonal parameters approach zero, this behavior has the side effect of also increasing the determinantal volumes of subsets within disjoint groups, since these volumes are also affected by these small parameter values. In



Figure 1: Results for synthetic experiment showing model recovery of structure of positive examples for low-sparsity data.



Figure 2: Results for synthetic experiment showing model recovery of structure of negative examples for high-sparsity data. 14 disjoint groups are used for data generation.

contrast, the nonsymmetric model behaves differently; determinantal volumes can be maximized by switching the signs of the off-diagonal entries of $K$ and increasing the magnitude of these parameters, rather than reducing the values of these parameters to near zero. This behavior allows the model to assign higher volumes to positive pairs than to negative pairs within disjoint groups in many cases, thus allowing the nonsymmetric model to recover disjoint structure.

In our experiment, the oracle controls the sparsity of the data by setting the number of disjoint groups of items; positive pairs within each disjoint group are generated uniformly at random, in order to focus on the effect of disjoint groups. For the AUC evaluation, negative baskets are constructed so that they contain items from at least two different disjoint groups. When constructing our dataset, we set the number of disjoint groups to 14, with 100 baskets of size six, and a catalog of 100 items. The results of our experiment are shown in Figure 2. We see from the error plot that the symmetric model cannot effectively learn the structure of the data, leading to high error in many areas, including within the disjoint blocks; the symmetric model provides an AUC of 0.5 as a result. In contrast, the nonsymmetric model is able to approximately recover the block structure, resulting in an AUC of 0.7.

**Recovering positive examples for data that mixes disjoint sparsity with popularity-based positive structure** For our final synthetic experiment, we construct a scenario that combines aspects of our two previous experiments. In this experiment, we consider three disjoint groups. For each disjoint group we use a categorical distribution with nonuniform event probabilities for sampling items within baskets, which induces a positive correlation structure within each group. Therefore, the oracle will expect to see a high negative correlation for disjoint pairs, compared to all other non-disjoint pairs within a particular disjoint group. For items with a high co-occurrence probability, we expect the symmetric DPP to recover a near zero negative correlation, and the nonsymmetric DPP to recover a positive correlation. Furthermore, we expect both the nonsymmetric and symmetric models to recover higher marginal probabilities, or $K_{ii}$ values, for more popular items. The determinantal volumes for positive pairs containing popular items will thus tend to be larger than the volumes of negative pairs. Therefore, for baskets containing popular items, we expect that both the nonsymmetric and symmetric models will be able to easily discriminate between positive and negative baskets. When constructing positive baskets, popular items are sampled with high probability, proportional to their popularity. We therefore expect that both models will be able to recover some signal about the correlation structure of the data within each disjoint group, resulting in a predictive AUC higher than 0.5, since the popularity-based positive correlation structure within each group allows the model to recover some structure about correlations among item pairs within each group. However, we expect that the nonsymmetric model will provide better predictive performance than the symmetric model, since its properties enable recovery of disjoint structure (as discussed previously). We see the expected results in Figure 3, which are further confirmed by the predictive AUC results: 0.7 for the symmetric model, and 0.75 for the nonsymmetric model.

## 5.4 Results on real-world datasets

To examine how the nonsymmetric model behaves when trained on a real-world dataset with clear disjoint structure, we first train and evaluate the model on the three-category Amazon baby registry

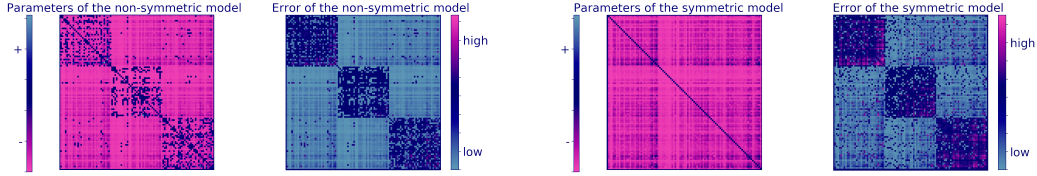

Figure 3: Results for synthetic experiment showing model recovery of positive structure for data with popularity-based positive examples and disjoint groups. Three disjoint sets and popularity-based weighted random generation are used for the positive examples.

| Metric | Amazon: Apparel | | Amazon: 3-category | | UK Retail | |
|--------|--------|--------|--------|--------|--------|--------|
| | Sym DPP | Nonsym DPP | Sym DPP | Nonsym DPP | Sym DPP | Nonsym DPP |
| MPR | $77.42 \pm 1.12$ | $\mathbf{80.32 \pm 0.75}$ | $60.61 \pm 0.94$ | $\mathbf{75.09 \pm 0.85}$ | $76.79 \pm 0.60$ | $\mathbf{79.45 \pm 0.57}$ |
| AUC | $0.66 \pm 0.01$ | $\mathbf{0.73 \pm 0.01}$ | $0.70 \pm 0.01$ | $\mathbf{0.79 \pm 0.01}$ | $0.57 \pm 0.001$ | $\mathbf{0.65 \pm 0.01}$ |

Table 1: MPR and AUC results for the Amazon Diaper, Amazon three-category (Apparel + Diaper + Feeding), and UK retail datasets. Results show mean and 95% confidence estimates obtained using bootstrapping. Bold values indicate improvement over the symmetric low-rank DPP outside of the confidence interval. We use $D = 30, \alpha = 0$ for both Amazon datasets; $D' = 100$ for the Amazon 3-category dataset; $D' = 30$ for the Amazon apparel dataset; $D = 100, D' = 20, \alpha = 1$ for the UK dataset; and $\beta = \gamma = 0$ for all datasets.

dataset. This dataset is composed of three disjoint categories of items, where each disjoint category is composed of 100 items and approximately 10,000 baskets. Given the structure of this dataset, with a small item catalog for each category and a large number of baskets relative to the size of the catalog, we would expect a relatively high density of positive pairwise item correlations within each category. Furthermore, since each category is disjoint, we would expect the model to recover a low density of positive correlations between pairs of items that are disjoint, since these items do not co-occur within observed baskets. We see the experimental results for this three-category dataset in Figure 4. As expected, positive correlations dominate within each category, e.g., within category 1, the model encodes 80.5% of the pairwise interactions as positive correlations. For pairwise interactions between items within two disjoint categories, we see that negative correlations dominate, e.g., between $C_1$ and $C_2$, the model encodes 97.2% of the pairwise interactions as negative correlations (or equivalently, 2.8% as positive interactions).

Table 1 shows the results of our performance evaluation on the Amazon and UK datasets. Compared to the symmetric DPP, we see that the nonsymmetric DPP provides moderate to large improvements on both the MPR and AUC metrics for all datasets. In particular, we see a substantial improvement on the three-category Amazon dataset, providing further evidence that the nonsymmetric DPP is far more effective than the symmetric DPP at recovering the structure of data that contains large disjoint components.

$$
\begin{array}{cccc}
 & C_1 & C_2 & C_3 \\
C_1 & \begin{bmatrix} \mathbf{80.5\%} & 2.8\% & 3.7\% \\ C_2 & 2.8\% & \mathbf{71.6\%} & 4.5\% \\ C_3 & 3.7\% & 4.5\% & \mathbf{97.8\%} \end{bmatrix}
\end{array}
$$

Figure 4: Percentage of positive pairwise correlations encoded by nonsymmetric DPP when trained on the three-category Amazon baby registry dataset, as a fraction of all possible pairwise correlations. Category $n$ is denoted by $C_n$.

## 6 Conclusion

By leveraging a low-rank decomposition of the nonsymmetric DPP kernel, we have introduced a tractable MLE-based algorithm for learning nonsymmetric DPPs from data. To the best of our knowledge, this is the first MLE-based learning algorithm for nonsymmetric DPPs. A general framework for the theoretical analysis of the properties of the maximum likelihood estimator for a somewhat restricted class of nonsymmetric DPPs reveals that this estimator has certain statistical guarantees regarding its consistency. While symmetric DPPs are limited to capturing only repulsive item interactions, nonsymmetric DPPs allow for both repulsive and attractive item interactions, which lead to fundamental changes in model behavior. Through an extensive experimental evaluation on several synthetic and real-world datasets, we have demonstrated that nonsymmetric DPPs can provide significant improvements in modeling power, and predictive performance, compared to symmetric DPPs. We believe that our contributions open to the door to an array of future work on nonsymmetric DPPs, including an investigation of sampling algorithms, reductions in computational complexity for learning, and further theoretical understanding of the properties of the model.

## Footnotes

*Currently at Google.

[2] Our code is available at https://github.com/cgartrel/nonsymmetric-DPP-learning

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
