[Supplementary Material]

# A Counterexamples to the backward implication in Part 3 of Lemma 3

Here, under nonsymmetric scenarios, we provide some counterexamples that show that (10) can be satisfied for nonzero matrices $\boldsymbol{H} \in \mathcal{H}$ even though $\boldsymbol{L}^*$ is not block diagonal. This implies that irreducible (i.e., not block diagonal up to relabeling of the items) matrices may be significantly harder to learn in the nonsymmetric case.

**Case of signed $P$-matrices with unknown signs**   Let $\varepsilon = \pm 1$ and define

$$\boldsymbol{L}^* = \begin{pmatrix} 1 & 1/2 \\ \varepsilon/2 & 1 \end{pmatrix}$$

and

$$\boldsymbol{H} = \begin{pmatrix} 0 & 1 \\ -\varepsilon & 0 \end{pmatrix}.$$

Then, $\boldsymbol{H} \neq 0$, it satisfies (10) and yet, $\boldsymbol{L}^*$ is irreducible.

In the $3 \times 3$ case, Define

$$\boldsymbol{L}^* = \begin{pmatrix} 1 & 1/2 & 0 \\ 1/2 & 1 & 1/2 \\ 0 & 1/2 & 1 \end{pmatrix}$$

and

$$\boldsymbol{H} = \begin{pmatrix} 0 & 0 & 1 \\ 0 & 0 & 0 \\ -1 & 0 & 0 \end{pmatrix}.$$

Then, $\boldsymbol{H} \neq 0$, it satisfies (10) and yet, $\boldsymbol{L}^*$ is irreducible. Note that $\boldsymbol{L}^*$ is symmetric. However, since we are under a scenario where it is not known beforehand that $\boldsymbol{L}^*$ is symmetric, the space of perturbations $\mathcal{H}$ is much larger.

**Case of matrices of the form $D + A$, where $D$ is diagonal with positive entries and $A$ is skew-symmetric**   Let

$$\boldsymbol{L}^* = \begin{pmatrix} 1 & 1/2 & 0 \\ -1/2 & 1 & 1/2 \\ 0 & -1/2 & 1 \end{pmatrix}$$

and

$$\boldsymbol{H} = \begin{pmatrix} 0 & 0 & 1 \\ 0 & 0 & 0 \\ -1 & 0 & 0 \end{pmatrix}.$$

Again, we see that $\boldsymbol{H} \in \mathcal{H}$ is nonzero, it satisfies (10) and yet, $\boldsymbol{L}^*$ is irreducible.

The case of matrices that are the sum of a positive diagonal matrix and a skew-symmetric matrix is particularly interesting, in applications where only attractive interactions between items (i.e., nonnegative correlations) are sought for. In this case, it would be interesting to be able to characterize the nullspace of the Fisher information, as in [6, Theorem 3].

**Open Question.** Let $\Theta$ be the set of all matrices $\boldsymbol{L} = \boldsymbol{D} + \boldsymbol{A}$, where $\boldsymbol{D}$ is a positive diagonal matrix and $\boldsymbol{A}$ is skew-symmetric. Recall that by Lemma 2, $\mathcal{H}$ is the set of all matrices of the form $\boldsymbol{D} + \boldsymbol{A}$ where $\boldsymbol{D}$ is any diagonal matrix and $\boldsymbol{A}$ is any skew-symmetric matrix.

1. Characterize the set of all $\boldsymbol{L}^* \in \Theta$ such that $\boldsymbol{H} = 0$ is the only solution in $\mathcal{H}$ to (10).

2. For a given $\boldsymbol{L}^* \in \Theta$, characterize the set of all solutions $\boldsymbol{H} \in \mathcal{H}$ of (10).

# B Proofs

**Proof of Lemma 1**   By the generating method 4.2 in [28], if $\boldsymbol{L} + \boldsymbol{L}^\top$ is positive definite, then $\boldsymbol{L}$ is a $P$-matrix, hence, it is a $P_0$-matrix. Assume now that $\boldsymbol{L} + \boldsymbol{L}^\top$ is only PSD. Let $\varepsilon > 0$ and $\boldsymbol{L}_\varepsilon = \boldsymbol{L} + \varepsilon \boldsymbol{I}$. Then, $\boldsymbol{L}_\varepsilon + \boldsymbol{L}_\varepsilon^\top = \boldsymbol{L} + \boldsymbol{L}^\top + 2\varepsilon \boldsymbol{I}$ is positive definite, hence, $\boldsymbol{L}_\varepsilon$ is a $P$-matrix, by the argument given above, so it is a $P_0$-matrix. Let $\xi(\boldsymbol{X}) = \min_{J \subseteq [M]} \det \boldsymbol{X}_J$, for $\boldsymbol{X} \in \boldsymbol{R}^{M \times M}$. Then, $\xi$ is a continuous function and the set of $P_0$-matrices is the set of all matrices $\boldsymbol{X}$ with $\xi(\boldsymbol{X}) \geq 0$, hence, it is closed. Therefore, $\boldsymbol{L}$ is a $P_0$-matrix, as the limit of the $P_0$-matrix $\boldsymbol{L}_\varepsilon$ as $\varepsilon \to 0$.

**Proof of Lemma 2** We only prove the third statement, since the first and the fifth ones are very simple, and the second and the fourth ones are directly implied by the third. For $i, j \in [M]$, we let $\boldsymbol{E}_{i,j}$ be the elementary $M \times M$ matrix with zeros everywhere but at the $(i, j)$-th entry, where we put a one.

If $i = j$, then $\boldsymbol{I} + \boldsymbol{E}_{i,i}$ and $\boldsymbol{I}$ are both in $\Theta$, hence, their difference $\boldsymbol{E}_{i,i}$ is in $\mathcal{H}$. If $i \neq j$, note that $2\boldsymbol{I} + (\boldsymbol{E}_{j,i} + \boldsymbol{E}_{i,j})$ and $2\boldsymbol{I} + (\boldsymbol{E}_{j,i} - \boldsymbol{E}_{i,j})$ are both in $\Theta$, since they are diagonally dominant. Hence, their difference $2\boldsymbol{E}_{i,j}$ is in $\mathcal{H}$, yielding $\boldsymbol{E}_{i,j} \in \mathcal{H}$. Therefore, $\boldsymbol{E}_{i,j} \in \mathcal{H}$ for all $i, j \in [M]$, yielding $\boldsymbol{R}^{M \times M} \subseteq \mathcal{H}$, since all matrices are linear combinations of the elementary matrices.

**Proof of Theorem 1** The arguments are almost identical to the ones given in [6, Theorem 2]. The main idea is to note that for all matrices $\boldsymbol{L} \in \boldsymbol{R}^{M \times M}$,

$$\det(\boldsymbol{I} + \boldsymbol{L}) = \sum_{J \subseteq [M]} \det \boldsymbol{L}_J, \tag{13}$$

which is a consequence of the $M$-linearity of the determinant. Then, we differentiate (13) twice on the set of $P$-matrices, by noticing that this is an open set. Indeed, recalling the notation $\xi$ from the proof of Lemma 1, the set of $P$-matrices is the set of all matrices $\boldsymbol{L}$ such that $\xi(\boldsymbol{L}) > 0$, and $\xi$ is continuous. Hence, following the same computations as in the proof of [6, Theorem 2] yields the desired result.

**Proof of Lemma 3**

1. Take $J = \{i\}$ in (10) for some $i \in [M]$. Then, $\boldsymbol{L}_J^*$ is a $1 \times 1$ matrix, whose single entry is the $i$-th diagonal entry of $\boldsymbol{L}^*$. Since $\boldsymbol{L}^* \in P$, $L_{i,i}^* \neq 0$, yielding that $H_{i,i}$ must be zero.

2. Now, take $J = \{i, j\}$ and write $\boldsymbol{L}_J^* = \begin{pmatrix} a & b \\ \varepsilon b & c \end{pmatrix}$, where $a$ and $c$ are nonzero since $\boldsymbol{L}^*$ is a $P$-matrix and $b = L_{i,j}^* \neq 0$ by assumption. Using the first part of the Lemma, it must hold that $H_{i,i} = H_{j,j} = 0$, hence, we write $\boldsymbol{H}_J = \begin{pmatrix} 0 & h \\ \varepsilon h & 0 \end{pmatrix}$ for some $h \in \boldsymbol{R}$. A direct computation shows that if (10) is satisfied, then $h$ must be zero.

# C  Mean Percentile Rank

We begin our definition of MPR by defining percentile rank (PR). First, given a set $J$, let $p_{i,J} = \Pr(J \cup \{i\} \mid J)$. The percentile rank of an item $i$ given a set $J$ is defined as

$$\mathrm{PR}_{i,J} = \frac{\sum_{i' \notin J} \mathbb{1}(p_{i,J} \geq p_{i',J})}{|\mathcal{Y} \backslash J|} \times 100\%$$

where $\mathcal{Y} \backslash J$ indicates those elements in the ground set $\mathcal{Y}$ that are not found in $J$.

MPR is then computed as

$$\mathrm{MPR} = \frac{1}{|\mathcal{T}|} \sum_{J \in \mathcal{T}} \mathrm{PR}_{i, J \backslash \{i\}}$$

where $\mathcal{T}$ is the set of test instances and $i$ is a randomly selected element in each set $J$.