[Reviews · NeurIPS 2019]

Reviewer 1



The developments presented here for dealing with asymmetric DPPs make this class of model more useful when positive correlations need to be modeled in the sampled sets, making the models more powerful than traditional symmetric DPPs. The model, theory and experiments are clearly explained. Experimental results show the possible improvements provided by the nonsymmetric DPPs which can model positive correlations as well as their ability to learn negative correlations as is the case in traditional symmetric DPPs. Experiments on real data further validate the advantage of nonsymmetric DPPs in both MPR and AUC measures for recommended sets.

Reviewer 2



As mentioned in the previous section, this work proposes a solution to an important and interesting problem and the empirical results are convincing enough to establish this approach over the symmetric DPP approach for the tasks and datasets described in the paper. One major thing missing is the lack of other baselines apart from symmetric DPPs. What about simpler models for set selection like tasks which are less computationally intensive; these would establish the necessity for DPPs for the selected task. More egregiously, how does this approach compare to the one on learning signed DPPs (Brunel 2017), which has been cited but not compared against. The final method in the paper looks different enough to merit a comparison.

Reviewer 3



This paper studies determinantal point processes (DPP) with non-symmetric kernels. Most of the machine learning literature on DPP assumes symmetric kernels, and the prior work that studies non-symmetric kernels have assumed a quite restricted class of non-symmetric kernels. The novelty of this paper is in proposing the learning algorithm for a fairly general class of non-symmetric kernels. The proposed approach assumes a particular representation of non-symmetric kernels. This representation follows from two known results in a rather straightforward manner, as I also summarize in "1. Conclusion". Once this representation is given, the algorithm follows essentially in the same way as Gartrell+ 2017. Nevertheless, the proposed representation is new in the particular context of learning the kernels for DPP. The proposed approach can fail when the expression in logdet() of the first term of the right-hand side of Eq. (12) is singular, which can be the case when D < M (low rank). The proposed representation assumes low rank, but the property of low rank is not quite exploited in the proposed algorithm. The low rank representation can obviously reduce the space complexity, as is claimed in the paper, but is this relevant in practice? The time complexity seems to be the bottleneck in practice. Given these observations, the low rank representation only creates a pitfall (singularity in (12)) without much benefit (time complexity not improved). --- I thank the authors for the clarification regarding the subclass of P0-matrics. I now find that the proposed representation is not obvious, and this representation itself is the major contribution of the paper. Regarding the time complexity, I see that the low-rank representation reduces the time for matrix multiplication, but this is not the bottleneck. The bottleneck remains O(M^3) time. However, the time complexity is a secondary issue and should not be a reason to reject this paper. The only remaining major issue is the possible singularity of the first term of Eq. 12. In the response, the authors suggest a heuristic to deal with singularity by adding "epsilon I". This additional term is essential and must be stated in the paper. In my opinion, it should be added into Eq. 12, because otherwise the proposed algorithm can fail. Overall I like the paper despite the deficiency.

[Author Response · NeurIPS 2019]

We thank the reviewers for their insightful comments and detailed analysis of our work. We provide clarifications below to address their comments.

**Reviewer 1** We thank this reviewer for a thoughtful discussion of our work, and we hope that our comments to other reviewers below will be helpful in clarifying our contributions.

**Reviewer 2**

− *Comparison to signed DPPs:* There are several important differences between our work and the work on learning signed DPPs [1]: **(1)** Signed DPPs require $K_{ij} = \pm K_{ji}$, where $K$ is the marginal DPP kernel (described in Sec. 2.1 of our paper). In contrast, our nonsymmetric DPP model is much more general, since it does not require that $|K_{ij}| = |K_{ji}|$. Since the off-diagonal elements of $K$ determine the correlations between pairs of items, $\mathrm{cov}(\mathbb{1}_{i \in Y}, \mathbb{1}_{j \in Y}) = -K_{ij}K_{ji}$, this gives our model more flexibility. **(2)** The learning algorithm for signed DPPs presented in [1] assumes that the unknown kernel $K$ is dense, i.e., all its entries are nonzero. In practice, this may not be a realistic assumption, because it implies that all pairs of items are correlated. **(3)** Moreover, our approach allows us to leverage a low-rank assumption on $L$ (or, equivalently, $K$), whereas the approach in [1] is not compatible with a low-rank assumption. **(4)** The learning algorithm in [1] has computational complexity of $O(M^6)$, where $M$ is the size of the ground set (e.g., item catalog), making it computationally infeasible for most scenarios. In contrast, our learning algorithm has substantially lower time complexity, which allows our approach to be used on many real-world datasets. It is true that [1] inspired and informed our work. We will add some text to the camera-ready version of our paper to provide a comparison with this work.

− *Comparison to other baselines:* Since the focus of our work is on improving DPP modeling power and comparing nonsymmetric and symmetric DPPs, to keep the message of our paper clear we use the standard symmetric low-rank DPP as the baseline model for our experiments. We plan to perform an experimental comparison to other competing models for subset selection as part of future work. Regarding the comparison with the theory presented in [2], we emphasize, in our work, that the problem becomes significantly harder when we deal with nonsymmetric kernels, which shows that going from symmetric to nonsymmetric kernels is not a straightforward extension of previous work.

**Reviewer 3**

− *Low-rank representation of nonsymmetric DPP kernel:* The first term on the right side of Eq. 12 will be singular whenever $|Y_i| > D$, where $Y_i$ is an observed subset. Therefore, to address this in practice we set $D$ to the size of the largest subset observed in the data, as explained in [3]. Furthermore, the first term on the right side of Eq. 12 may be singular even when $|Y_i| \leq D$. In this case, we know that we are not at a maximum, since the value of the function becomes $-\infty$. Numerically, to prevent such singularities, in our implementation we add a small $\epsilon I$ correction to each $L_{Y_i}$ when optimizing Eq. 12 (we set $\epsilon = 10^{-5}$ in our experiments).

Regarding the significance of our low-rank decomposition of $L$ for nonsymmetric DPPs (described in lines 177 - 181 of our paper), this is indeed an extension of an idea developed in the symmetric case, and we do use well known decompositions for symmetric and skew symmetric matrices. We do not claim that we prove new matrix decompositions, but we rather propose a simple low-rank representation of a **subclass** of $P_0$-matrices. Please note that the claim, in the review, that a $P_0$-matrix can be decomposed as the sum of a PSD matrix and a skew-symmetric matrix is incorrect, and is not a consequence of Lemma 1 in our paper. Lemma 1 only states that if the symmetric component of a matrix is PSD, then that matrix is $P_0$, but the converse is not true (e.g., take the $P_0$-matrix $L = ((1, -1), (5, 1))$, whose symmetric component, $(L + L^T)/2$, is the non-PSD matrix $((1, 2), (2, 1))$). Therefore, dealing with the class of all $P_0$-matrices seems very challenging, but leaves an exciting research topic open.

Regarding the time complexity of the low-rank representation, we see from Eq. 12 that the time complexity required to compute the matrix multiplications associated with the gradient of the first and second terms of the log-likelihood will be $O(n\kappa^2 D + n\kappa^2 D' + DM^2 + D'M^2)$, where $n$ is the number of observed subsets, $\kappa$ is the size of the largest observed subset in the training data, and $M$ is the size of the ground set (item catalog). We typically set $D \ll M$ and $D' \ll M$ in the low-rank representation; the associated matrix multiplications become much more expensive if we set $D = M$ (and presumably $D' = M$). In particular, the matrix multiplications for the second term of the log-likelihood will become $O(M^3)$ operations, instead of $O(DM^2 + D'M^2)$ operations. Therefore, we see that our low-rank representation still affords improvements in time complexity compared to the full-rank representation. We will add some text to the camera-ready version of our paper to make this point clear. We are confident that it is possible to approximate the DPP normalization constant, $\log \det(L + I)$, using contrastive estimation for DPPs [4], and therefore address the remaining $O(M^3)$ bottleneck, but we leave this for future work.

# References

[1] Victor-Emmanuel Brunel. Learning signed determinantal point processes through the principal minor assignment problem. In *NeurIPS*, pages 7365–7374, 2018.

[2] Victor-Emmanuel Brunel, Ankur Moitra, Philippe Rigollet, and John Urschel. Rates of estimation for determinantal point processes. In *Conference on Learning Theory*, pages 343–345, 2017.

[3] Mike Gartrell, Ulrich Paquet, and Noam Koenigstein. Low-rank factorization of Determinantal Point Processes. In *AAAI*, 2017.

[4] Zelda Mariet, Mike Gartrell, and Suvrit Sra. Learning determinantal point processes by corrective negative sampling. In *AISTATS*, pages 2251–2260, 2019.


[Meta-Review · NeurIPS 2019]

All reviewers and myself think that the proposed representation is nontrivial and useful. One reviewer still has some concern about the cubic computational complexity, but it's not a major concern. Overall this is a solid paper and deserves publication.